# The Question of a Role for Statins in Age-Related Macular Degeneration

**DOI:** 10.3390/ijms19113688

**Published:** 2018-11-21

**Authors:** Marina Roizenblatt, Nara Naranjit, Mauricio Maia, Peter L. Gehlbach

**Affiliations:** 1Department of Ophthalmology, Johns Hopkins University School of Medicine, Baltimore, MD 21287, USA; maroizenb@gmail.com (M.R.); nnaranj2@jhmi.edu (N.N.); 2Department of Ophthalmology, Federal University of São Paulo, São Paulo 04023-062, Brazil; maiamauricio@terra.com.br; 3Vision Institute, IPEPO, Department of Ophthalmology, Paulista Medical School, Federal University of São Paulo, São Paulo 04038-032, Brazil

**Keywords:** age-related macular degeneration, atherosclerotic disease, cholesterol, retina, statins

## Abstract

Age-related macular degeneration (AMD) is the leading cause of irreversible central vision loss in patients over the age of 65 years in industrialized countries. Epidemiologic studies suggest that high dietary fat intake is a risk factor for the development and progression of both vascular and retinal disease. These, and other associations, suggest a hypothesis linking elevated cholesterol and AMD progression. It follows, therefore, that cholesterol-lowering medications, such as statins, may influence the onset and progression of AMD. However, the findings have been inconclusive as to whether statins play a role in AMD. Due to the significant public health implications of a potential inhibitory effect of statins on the onset and progression of AMD, it is important to continually evaluate emerging findings germane to this question.

## 1. Introduction

As a consequence of an aging population, age-related macular degeneration (AMD) has become a global issue. An estimated 1.8 million individuals in the United States have either geographic atrophy or neovascular AMD in at least one eye, and this number is expected to increase to nearly 3 million by 2020 [1]. Significant resources are therefore being directed toward understanding the role of genetics and modifiable risk factors and identifying new therapeutic options [2,3].

In addition to being the leading cause of blindness in the elderly in the developed world, AMD is the third leading cause of blindness worldwide, accounting for 8.7% of all cases of permanent vision loss [4,5]. AMD can be broadly classified into non-neovascular and neovascular forms. The non-neovascular stage is characterized by drusen, retinal pigment epithelium (RPE) degeneration, and atrophy of photoreceptors. The more advanced neovascular stage is characterized by the development of choroidal neovascularization [6]. Although neovascular AMD represents only 10% of the disease burden, it is responsible for 90% of AMD-related blindness [7].

Ischemia is among the contributing factors to the transition from non-neovascular to neovascular disease, as well as progression to the late-stage disease. Further supporting an ischemic hypothesis are studies proposing that AMD represents a pathologic tissue response to the damage caused by local ischemic insult. This, in turn, leads to elevated levels of vascular endothelial growth factor (VEGF) and choroidal neovascularization [8,9,10].

VEGF has emerged as a key regulator of pathologic angiogenesis and is, at present, the most common target in the treatment of choroidal neovascularization. Following intravitreous injection, anti-VEGF drugs block abnormal retinal blood vessel growth and decrease vascular permeability, thereby decreasing vascular leakage. Decreased intraretinal and subretinal fluid in the macula is in turn associated with improvement in the patient’s central vision [11,12]. The implementation of novel therapeutic approaches, based on our present incomplete knowledge, is ever changing and evolving.

Given the paucity of treatment for dry disease, efforts to better understand its pathophysiology and to develop relevant model systems continue. Model systems have historically led to significant breakthroughs in other fields, such as cardiovascular disease and neurodegeneration. In AMD, much of the pertinent pathology is not revealed by color fundus photography, which has historically played a major role in staging and following the progression of disease. This has led to a variety of diagnostic tools being used in major epidemiology and genome-wide association studies [13].

Of the modifiable risk factors, smoking is considered the most significant, and it is responsible for an approximately 3–4-fold increased risk of disease among current smokers [14]. A number of other factors may also be involved in the pathogenesis of AMD, including, but not limited to: aging, genetic markers, alcohol, exposure to sunlight, nutritional factors, and cardiovascular disease [7]. Various nutritional factors are implicated in the pathogenesis of AMD. The Age-Related Eye Disease Study (AREDS) clinical trial evaluated an antioxidant vitamin formulation and demonstrated its benefit in a subgroup of non-neovascular AMD patients. The actual benefit demonstrated was a reduction in the risk of conversion from non-neovascular to neovascular disease [15,16]. In addition to smoking, AMD and cardiovascular disease share a number of described risk factors and associations, including, but not limited to, obesity, hypertension, atherosclerosis, and stroke [17]. If this hypothesis is born out, the consumption of a number of dietary constituents—most notably, high dietary fat—might be a risk factor for the development and progression of both coronary and retinal microvascular disease, as well as AMD [18,19].

The described parallels between AMD and atherosclerotic disease do predict a relationship between the use of cholesterol-lowering medications and AMD. A number of clinical studies have now examined this possibility. The hydroxymethylglutaryl (HMG)-CoA reductase inhibitors (statins) are now a first-line therapy for the prevention of cardiovascular disease, nonfatal cardiovascular events, and mortality. Statins are used worldwide in patients with hypercholesterolemia and coronary artery disease, as they decrease endogenous cholesterol synthesis and consequently lower the serum levels of low-density lipoprotein (LDL) cholesterol [20,21]. As secondary actions, statins exert anti-inflammatory, antithrombotic, and antioxidant effects, as well as increase nitric oxide production and improve vascular endothelial cell function [22]. Any of the primary, secondary, or combined actions of statins could be beneficial in AMD. Therefore, the hypothesis that statins could benefit AMD remains plausible. As with the impact of the AREDS, vitamin formulation, even a small improvement, would have a great effect on vision, patient care, cost of care, and patient quality of life [17,23].

## 2. Serum Cholesterol and AMD (Age-Related Macular Degeneration)

The retina has multiple physiological demands for cholesterol utilization. The relative contribution to these needs by exogenous LDL-derived cholesterol and endogenously synthesized cholesterol remains unknown [24,25]. Cellular cholesterol levels represent a balance of endogenous synthesis and the uptake of exogenous cholesterol (mainly delivered by LDL in the systemic circulation) on one hand, and turnover via release, on the other [26]. Most cells have the capacity for synthesis of cholesterol, but few (liver, intestine, heart, kidney, RPE) have the capacity for assembly and secretion of lipoproteins [27,28]. The retina is a metabolically active tissue and subject to outer segment shedding. The cholesterol used in membrane synthesis is, in part, derived from hepatic sources via the blood, given the slow rate of lipid synthesis within the retina and the slow turnover in the lipid from photoreceptor outer segments [29]. Rodriguez et al. published a report on localizing apolipoproteins and receptors in primates and provide evidence for the rapid turnover of retinal cholesterol [30,31]. More recently, Pikuleva published important data about cholesterol homeostasis in mouse retina [28,32,33]

There are physiological changes in cholesterol metabolism that occur with aging, and these affect the RPE [34,35]. Drusen and basal linear deposit are quite specific for AMD, and cholesterol is present in drusen, as described in several studies using specific histochemistry [36,37,38,39,40] (Figure 1). Basal laminar deposits occur in several monogenic inherited disorders (Sorsby, TIMP3; Malattia Leventinese, FIB3; late-onset retinal degeneration, C1QTNF5; and others), in addition to being a strong risk factor for AMD progression when progression involves the macula/fovea [41]. AMD is, however, regarded as a multifactorial disease affecting the photoreceptors and their support system, including the RPE–Bruch membrane (BM) complex. Drusen are pathognomonic disease markers both on clinical examination and in histopathological study [42]. Drusen are focal, dome-shaped extracellular lesions between the RPE basal lamina and the inner collagenous layer of the BM [27,43]. The observation that lipid particles accumulate within the BM in the same location, prior to the development of basal deposits or drusen, has led to the hypothesis that cholesterol contributes to drusen formation during the development of AMD [27,36,44,45].

Early studies [46,47] have suggested that hyperlipidemia/hypercholesterolemia might be a risk factor for AMD progression, but, at this time, the consensus of the literature does not strongly support this conclusion. Eye pathology studies demonstrate a high cholesterol concentration in classical AMD lesions, such as drusen, aging BM, and newly discovered subretinal lesions. In addition, the association between several variants in cholesterol-related genes and AMD suggests that these variants may play important roles in early AMD [48,49,50,51]. Mendelian randomization studies [52] show that the strength of the lipid association in AMD is high, and the association is intermediate between cardiovascular disease and Alzheimer’s disease. Pathology [42] and genetics [52] studies alike conclude that intraocular expression of cholesterol-handling genes could imply that plasma cholesterol levels are unrelated or even opposite to those in cardiovascular disease. Moreover, gene expression studies have documented that numerous cholesterol and lipoprotein-related genes are expressed in human RPE and retina [53,54,55]. Local expression is potentially important, as is expression in liver and intestines, which regulate plasma cholesterol levels

A high-fat diet in mice induces functional changes in the retina, ultimately impacting the underlying RPE and BM morphology [56]. Epidemiologic, genetic, and pathologic evidence further raises the possibility that statins may impact AMD development, and this relationship has been examined in a number of clinical studies.

There are a number of mechanisms by which statins may exert protective effects in AMD. These include, but are not limited to, serum lipid-lowering that may alter BM lipid deposition [57]; preservation of the outer retinal and choroidal vascular supply by an anti-atherogenic effect [58]; anti-inflammatory properties [58]; antioxidant effects that may counter increased plasma levels of oxidized LDL [59]; and inhibition of metalloproteinases that may contribute to fissuring and rupture of plaques that lead to neovascularization [60]. HMG-CoA reductase inhibition may also have direct effects on cholesterol processing by outer retinal cells. The RPE is a native secretor of lipoproteins, and statins may affect lipidation of lipoproteins directly [61].

Tian et al. showed that high doses of atorvastatin preserved the phagocytic function of human RPE cells by increasing cell membrane fluidity. Interestingly, atorvastatin not only increased the baseline phagocytic function of RPE cells, but also preserved the phagocytic function impaired by cholesterol crystals. Similar effects were observed while assessing lovastatin and simvastatin. Furthermore, atorvastatin also exhibited anti-inflammatory properties, blocking the induction of IL-6 and IL-8 interleukins triggered by pathological stimuli relevant to AMD. Both characteristics reinforce the role of statins as potentially effective drugs in the prevention and treatment of AMD by protecting RPE cells from the impairment of the phagocytic function and the inflammatory effects induced by cholesterol crystals [62].

## 3. Statins and AMD

The majority of effective AMD treatment options presently available are limited to the exudative stages of the disease [60]. For non-neovascular AMD, the AREDS vitamin formulation is the only medical intervention for which there is level one evidence of prevention of progression of disease [63]. Numerous treatments have been used for neovascular AMD, including laser photocoagulation [64], transpupillary thermal therapy [65], photodynamic therapy [66], and various anti-VEGF injections [67]. However, repeated intravitreal injections of anti-VEGF drugs, such as ranibizumab (Lucentis), bevacizumab (Avastin), and aflibercept (Eylea), have become the standard of care for the treatment of neovascular AMD [67,68].

The administration of anti-VEGF medications requires multiple intravitreous injections for an indefinite duration. This mode of therapy negatively affects both the quality of life of affected patients and has a significant negative economic impact. Importantly, recovery or preservation of vision is not assured [69]. Subretinal and/or intraretinal fluid may persist or be recurrent in many patients, and such incomplete responses are treated with longer duration therapy, a change in anti-VEGF drugs, or combination therapy. Many large clinical trials have explored modified treatment regimens in an attempt to decrease the burden of treatment without compromising vision outcomes [70,71]. In addition, there is no effective treatment available for geographic atrophy or disciform scar formation [72].

Avoiding exudative disease and associated injection therapy would represent significant progress, but there have been few recent advances, underscoring the importance of assessing any potential statin-related benefit. Considering the limited therapeutic modalities available at present, there is no doubt that a new class of drugs, such as statins, would be able to expand the therapeutic arsenal and be of significant clinical relevance for the treatment of AMD. Subsequently, a number of studies have examined the relationship between AMD and statin effectiveness compared to other treatment options (Table 1), absence of treatment, or placebo, but the results remain mixed [73].

### 3.1. Literature Supporting the Positive Association between Statins and AMD

Similarities in risk factors and pathogenesis between AMD and atherosclerosis have engaged the interest of global scientific communities to explore the effects of statins on the incidence and progression of AMD. Observational studies have attempted to correlate the use of statins with the development of disease through the analysis of AMD patients’ past medical history and relevant literature review.

Barbosa et al. performed a cross-sectional study including 5604 participants older than 40 years. Patients were asked if they carried a diagnosis of AMD, if they used any type of statin agents without regarding the dose, and about their comorbidities and health-related behaviors, such as smoking. This study included both early and late AMD patients, and the eye in worse condition was used to assign the participant a grade. The survey found that individuals 68 years or older who were classified as long-term users of statins had less self-reported AMD after adjustment for confounding factors (OR, 0.64; 95% CI, 0.49–0.84; *p* = 0.002) [74]. Klein et al. took a similar questionnaire approach when examining the association of statin use with a 5-year incidence of AMD in a large population-based epidemiologic study. The cohort included 2780 otherwise healthy people 48–91 years old. The study found that those who started statins during the previous 5 years were 32% less likely to have soft drusen, 36% less likely to have large soft drusen, and 71% less likely to have late AMD over the follow-up period, as compared with patients who never took statins. The article had no data regarding which statin was used, the drug dosage, or the duration of use [75].

Fong et al. reinforced the positive association between any statin class, without regard to dose, and AMD in a case-controlled study of 719 patients who were older than 60 years and newly diagnosed with exudative AMD. The work showed a statin protective effect of 0.70 against neovascular AMD. Drug use information was obtained using computerized databases of newly diagnosed AMD cases and healthy controls who had been submitted to clinical examination by an ophthalmologist during that year [76]. Moreover, through a systematic search of databases of eligible published literature, Ma et al. found that statin use was protective for early and exudative AMD. In this review, statin use significantly reduced the early AMD risk by approximately 17% (RR, 0.83; 95% CI, 0.66–0.99). In the exudative stage, they also observed a significant protective association of statin use (RR, 0.90; 95% CI, 0.80–0.99). The authors did not specify the group of statins studied or the dosage administered [77].

Clinical trials comparing statin effects in delaying the onset and progression of AMD have been published. Vavvas et al. performed a pilot multicenter prospective clinical trial in a high-risk subgroup of AMD individuals in which they analyzed 26 patients with a diagnosis of AMD and the presence of many large, soft drusenoid deposits; the patients then received high-dose atorvastatin for 12 months. The group presented with the regression of drusen associated with vision gain (+3.3 letters, *p* = 0.06) in 10 patients. None of the study’s patients progressed to advanced neovascular AMD [78]. Similarly, Guymer et al. reported that simvastatin may slow progression of non-advanced AMD, especially for those with the CFH genotype CC (Y402H), a well-established disease risk factor. A double-masked randomized controlled study with 114 participants with either bilateral intermediate AMD or unilateral non-advanced AMD (advanced AMD in fellow eye) were prescribed simvastatin 40 mg/day or placebo and were allocated 1:1. The cumulative AMD progression rates were 70% in the placebo and 54% in the simvastatin group, and multivariable logistic regression analysis showed a significant 2-fold decrease in the risk of retinal disease progression in the simvastatin group: OR 0.43 (0.18–0.99), *p* = 0.047 [79].

### 3.2. Literature Supporting the Negative Association between Statin and AMD

By contrast, the majority of studies available in the current scientific literature report insufficient evidence to justify the use of statins to slow AMD progression and onset. There are several observational surveys that attempt to correlate statin use with an AMD therapeutic effect. Al-Holou et al. concluded that there was no statistically significant evidence that statins have a beneficial effect in slowing AMD progression or in preventing the disease from progressing to the late stage (hazard ratio, 1.08; 95% CI, 0.83–1.41; *p* = 0.56). In this study, age-adjusted proportional hazards regression models were performed to evaluate the association of statin use with progression to late AMD in 3791 participants of whom 1659 (43.8%) were already previous statin users. The extent to which prior statin use confounded the comparison is not known.

In addition, the study lacked specified dosages of statins and exact duration of treatment [80]. Individual statins are also known to have differing effects on retinal cell lipid metabolism.

Maguire et al. [81] performed a similar cohort study to evaluate the impact of statin use on the incidence of advanced AMD among 764 patients with bilateral large drusen. Patients were asked about their use of statins on the day of ophthalmological exams, regardless of the drug class or dose. Development of advanced AMD, choroidal neovascularization, and geographic atrophy were the established main outcome measures. The estimated risk ratio for eyes (95% CI) associated with statin use was 1.15 (0.87–1.52) for advanced AMD, 1.35 (0.99–1.83) for choroidal neovascularization, and 0.80 (0.46–1.39) for geographic atrophy; the study did not show a strong protective effect (risk ratio, ≤0.85) of statins on the development of advanced AMD among patients with bilateral large drusen.

Further studies, also based on database analysis, called into question the validity of the statin hypothesis. A large, national insurance claims database of 107,007 beneficiaries was reviewed by VanderBeek et al. to identify whether statin use was associated with the development of AMD or the progression from non-neovascular to neovascular disease. They concluded that statin use was not associated with the development of non-neovascular AMD (*p* > 0.05). However, statin use for more than 12 months was surprisingly associated with an increased risk for developing neovascular AMD (*p* < 0.005). Among those taking statins, only the ones with the highest lipid levels had an increased risk of developing neovascular AMD (*p* < 0.05). Therefore, they suggested that lipid status influences the relationship between statins and the risk of AMD [82]. The sufficiency of statin dose at the highest lipid levels remains a potential confounding factor for future study. Further confounding the question is whether statin protective properties are maintained at the higher doses required to control the highest lipid levels. In general, studies have not correlated statin-induced control of the various lipid profile components with AMD progression, identified a dose–response, compared the effects of various statins on AMD, or matched the severity of AMD with the severity of cardiovascular or neurodegenerative disease. Also of interest may be the evaluation of inflammatory biomarkers on statin therapy. Prospective studies about this subject are needed in the future.

Based on large database analysis, Shalev et al. examined a population-based retrospective cohort of 108,973 individuals from a large health organization in Israel. The organization’s central computerized database was used to collect information on incident AMD cases that were already on a statin. The mean proportion of days covered was calculated by dividing the quantity of statins dispensed by the total time interval from index date to death. The crude incidence density rate of AMD among patients in the lowest quintile of the proportion of days covered was comparable to that of patients in the highest quintile. Moreover, after adjustment for potential confounders, patients with persistent use of statins had a risk ratio of 0.99 (95% CI, 0.78–1.26) for AMD compared with patients in the lowest proportion of days covered. These results suggest that early reports of a strong protective effect of statins for AMD development may have been the result of a small study effect [83].

Moreover, Gehlbach et al. performed a meta-analysis on this question in the form of a Cochrane system-wide review that identified only two randomized controlled trials from the literature that met the Cochrane inclusion criteria, leaving 144 participants for assessment. The study compared statins with placebos in patients diagnosed as having the early stages of AMD. Standard Cochrane criteria judged the quality of the evidence to be low due to limitations in the trial design and insufficient outcome reporting and concluded that the currently available literature is insufficient to establish that statins have a role in preventing or delaying the onset or progression of AMD [73]. The study did not rule out the possibility that an effect was present.

Martini et al. were included in the above-mentioned Cochrane review. Martini et al. were responsible for conducting a randomized clinical trial in which 30 participants with early-stage AMD received simvastatin (20 mg/day) or placebo for 3 months. The conclusion was that there was no statistically significant difference between the simvastatin and the placebo therapy in visual acuity, as they showed a decimal visual acuity of 0.21 ± 0.56 in the simvastatin group and 0.19 ± 0.40 in the placebo group. Furthermore, lens and retina status were unchanged during and after the treatment period for both groups [84]. It is possible that the exceedingly brief duration of treatment, given the slow rate of disease progression in the natural history of AMD, was insufficient to detect any benefit, especially as visual acuity is generally well preserved in dry AMD prior to conversion to wet disease. 

## 4. Limitations

Although logical, a pathophysiological correlation between ingestion of statins and slowing progression, or even preventing AMD, has yet to be proven. This is despite several studies in the literature that have tried to prove such a theory in clinical practice. In total, the findings have not yet been decisive, either in support of or against the hypothesis. Faced with this obstacle, researchers must first explain why this important question remains unanswered. It is worth mentioning that there is no standardization of the AMD staging in the patients analyzed in the majority of the abovementioned studies. The heterogeneity of AMD suggests that, perhaps, the effects of statins vary by stage, reducing the development of drusen at the onset of AMD and having a further anti-inflammatory effect on late disease [77]. In addition, genetic analysis should also be performed to understand whether the genotype is capable of influencing a response to this therapeutic option.

Another relevant point is the lack of standardization of statin dosages or individual lipophilicity for each class of drug, considering that the combined registration of the use of low-potency and high-potency statins causes a methodological bias [85]. Statin use may only be effective if lipid levels are reduced. It is still unclear how much reduction is required and which lipid subspecies to target in order to impact the natural history of AMD, if any effect exists [86]. Finally, other weaknesses in the study design that might lead to insufficient evidence that statins could prevent AMD include the following: short duration of follow-up, lack of data from participants, insufficient sample power to detect significant differences between treatment groups, and the fact that people taking statins, in epidemiological studies, are more likely to have other confounders that cannot be fully adjusted, such as high-risk cardiovascular disease and a large number of other medications. Resolving these issues in future studies should lead to a reliable and definitive ruling on the efficacy of statins in AMD.

## 5. Conclusions

AMD has grown into a global health concern that negatively and severely impacts quality of life for the millions affected, as well as the overall cost of healthcare. At present, AMD is a progressive chronic disease for which there is no cure. In the Western world, under the best care, it remains a leading cause of visual impairment in the elderly. A hypothesis implicating hyperlipidemia/hypercholesterolemia as a risk factor in AMD has arisen from the observation that AMD and cardiovascular disease share a number of risk factors and pathophysiological pathways. Thus, attention to the possibility that cholesterol-lowering medications, such as statins, already known to be effective in reducing cardiovascular disease, might also be beneficial in delaying onset and/or progression of AMD. The statin hypothesis has since been evaluated in several clinical trials to assess risk reduction of disease onset and progression. However, the results remain mixed and, in rare instances, contradictory. While, at present, there is not sufficient reliable evidence to support the hypothesis that statins prevent or delay AMD progression, there is also not sufficient evidence to put the question to rest. Given the large potential benefit to patient quality of life and cost of care that even a modest protective effect would provide, further study is reasonable and warranted.

## Figures and Tables

**Figure 1 ijms-19-03688-f001:**
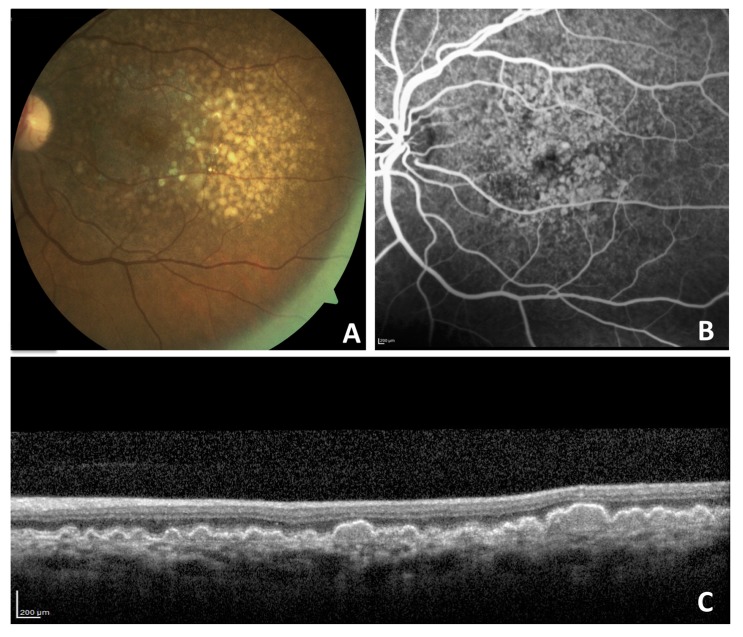
A 59-year-old woman with non-neovascular age-related macular degeneration with large confluent drusen on color fundus photograph (**A**), late-phase fluorescein angiography (**B**), and optical coherence tomography (**C**).

**Table 1 ijms-19-03688-t001:** Summary of findings supporting the positive or negative association between statins and AMD.

Reference	Statin group	Description of the Study	Scenario	Primary outcomes assessed	Subjects	Conclusion	Potential association
Tian et al., 2017 [62]	Atorvastatin 50 µM	The study describes the action of atorvastatin on the phagocytic function of ARPE-19 cells and on the inflammatory effects induced by crystals of cholesterol and ox-LDL	Non-clinical	Phagocytic function of ARPE-19 cells and induction of interleukins IL-6 and IL-8	ARPE-19 cell line	Statins help to preserve the phagocytic function of the RPE while also exhibiting anti-inflammatory properties	Positive
Barbosa et al., 2014 [74]	Not specified	Patients were asked if they carried a diagnosis of AMD, if they used any type of statin agents without regard to dose, and about their comorbidities and health-related behaviors, such as smoking.	Clinical	Correlated the number of long-term users of statins and the number of self-reported AMD patients after adjustment for confounding factors	5604 participants older than 40 years	Individuals of 68 years or older who were classified as long-term users of statins had less self-reported AMD after adjustment for confounding factors	Positive
Klein et al., 2003 [75]	Not specified	Questionnaire approach examining the association of statin use with a 5-year incidence of AMD in a large population-based epidemiologic study	Clinical	Chances of having soft drusen or late AMD among those who started statins during the previous 5 years as compared with patients who never took statins	2780 healthy people aged 48–91 years old	Those who started statins during the previous 5 years were less likely to have soft drusen and large soft drusen and less likely to have late AMD over the follow-up period	Positive
Fong et al., 2010 [76]	Not specified	Drug use information was obtained using computerized databases of newly diagnosed AMD cases and healthy controls who had seen an ophthalmologist during a period of one year	Clinical	Prevalence of neovascular AMD	719 patients older than 60 years newly diagnosed with exudative AMD	The work showed a statin protective effect against neovascular AMD	Positive
Ma et al., 2015 [77]	Not specified	A systematic search of the PubMed, EMBASE, and ISI web of science databases was used to identify eligible published literatures	Clinical	Evaluated the association between statin use and the risk of early and exudative AMD.	A total of 14 studies met the inclusion criteria and were included in this meta-analysis	For early AMD, statin use significantly reduced the risk. At the late stage, a significant protective association of statin use with exudative AMD, in contrast with the absent association between statins and geographic atrophy	Positive
Vavvas et al., 2016 [78]	80 mg of Atorvastatin	Patients with a diagnosis of AMD, the presence of many large, soft drusenoid deposits, and who then received high-dose atorvastatin for 12 months were evaluated	Clinical	Regression of drusen deposits, vision gained, and progression to advanced neovascular AMD	26 patients with a diagnosis of AMD	The group presented regression of drusen associated with vision gain in 10 patients. None of the study’s patients progressed to advanced neovascular AMD	Positive
Guymer et al., 2013 [79]	40 mg of Simvastatin	This was a 3-year study of simvastatin in participants with nonadvanced AMD in at least one eye, considered at high risk of progression toward advanced.	Clinical	Progression of AMD either to advanced AMD or in severity of non-advanced AMD	114 participants aged 53–91 years, with either bilateral intermediate AMD or unilateral non-advanced AMD (with advanced AMD in fellow eye), BCVA 20/60 in at least one eye, and a normal lipid profile	The cumulative AMD progression rates were higher in the placebo as compared to the simvastatin group	Positive
Al-Holou, 2015 [80]	Not specified	Age-adjusted proportional hazards regression models were used to evaluate the association of statin use with progression to late AMD	Clinical	Baseline and annual stereoscopic fundus photographs were assessed centrally for the development of late AMD, either neovascular AMD or geographic atrophy	3791 participants of whom 1659 were already previous statin users	There was no statistically significant evidence that statins had a beneficial effect in slowing AMD progression or in preventing the disease from progressing to the late stage	Negative
Maguire et al., 2009 [81]	Not specified	Patients were asked about their use of statins on the day of ophthalmological exams	Clinical	Development of advanced AMD, choroidal neovascularization, and endpoint geographic atrophy.	764 patients with bilateral large drusen	Statin use did not show a strong protective effect on the development of advanced AMD among patients with bilateral large drusen	Negative
VanderBeek et al., 2013 [82]	Not specified	Prescription for statins within a 24-month look-back period and outpatient lipid lab values were reviewed using an insurance database. Cox regression analysis was performed to determine whether statin use was associated with the development of non-exudative or exudative AMD, or progression from non-exudative to exudative AMD	Clinical	To determine if statins are associated with the development or progression of AMD	107,007 individuals aged ≥60 years who were enrolled for ≥2 years and had ≥1 visit(s) to an eye provider	In those with elevated lipid levels, >1 year of statin use was associated with an increased hazard for exudative AMD	Negative
Shalev et al., 2011 [83]	Not specified	The organization’s central computerized database was used to collect information on incident AMD cases that were already on a statin	Clinical	To investigate the association between persistent use of statins and the risk of AMD	108,973 individuals aged 55 or older who began statin therapy between 1998 and 2006 in a large health organization in Israel	The crude incidence density rate of AMD among patients in the lowest quintile of the proportion of days covered was comparable to that of patients in the highest quintile. Moreover, after adjustment for potential confounders, patients with persistent use of statins had a risk ratio for AMD comparable with patients in the lowest proportion of days covered	Negative
Gehlbach et al., 2016 [73]	20 or 40 mg of Simvastatin	Meta-analysis in the form of a Cochrane system-wide review that identified only two randomized controlled trials from the literature that met the Cochrane inclusion criteria.	Clinical	To examine the effectiveness of statins compared with other treatments, no treatment, or placebo in delaying the onset and progression of AMD	Randomized controlled trials and quasi-randomized trials that compared statins with other treatments, no treatment, or placebo in people who were diagnosed as having the early stages of AMD	Evidence from currently available randomized controlled trials is insufficient to conclude that statins have a role in preventing or delaying the onset or progression of AMD	Negative
Martini et al., 1991 [84]	20 mg of Simvastatin	Clinical trial in which participants with early-stage AMD received simvastatin or placebo for only 3 months	Clinical	Final visual acuity	30 participants with early-stage AMD	There was no difference between the simvastatin and the placebo therapy in terms of visual acuity	Negative

ox-LDL: oxidized low-density lipoproteins; AMD: age-related macular degeneration; BCVA: best corrected visual acuity; RPE: retinal pigment epithelium.

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
