# Peer review of "The Question of a Role for Statins in Age-Related Macular Degeneration"

_ijms, 2018, doi:10.3390/ijms19113688_

Round 1
Reviewer 1 Report
While the pathophysiology of AMD is not completely understood, there are well-established pathological changes, including extracellular lipoprotein and cholesterol-rich deposits between the retinal pigment epithelium (RPE) and Bruch’s membrane that seem to play a causal role in disease development and progression. Statins are known to reduce the endogenous synthesis of cholesterol and have been very successfully used as blood cholesterol-lowering agents. However, there are conflicting data regarding their benefits for AMD.
In this review Roizenblatt et al. evaluate the evidence, from observational surveys, clinical trials, database analysis, and Cochrane reviews, that supports positive and negative associations between Statin and AMD. Although the literature review presented here is interesting, and up to date, their analysis fails to provide any further insight on why the use of these drugs for AMD still remains inconclusive, in spite of many years of intense investigation. Also, the authors should propose what else should be done to reliably decide the effectiveness of Statins in AMD.
Author Response
A “limitation” section was included at the end of the main text, as follows:
4. LIMITATIONS
Although logical, a pathophysiological correlation between ingestion of statins and slowing progression, or even preventing AMD, has yet to be proven. This is despite several studies in the literature that have tried to prove such a theory in clinical practice. In total the findings have not yet been decisive, either in support of or against the hypothesis. Faced with this obstacle, researchers must first explain why this important question remains unanswered. It is worth mentioning that there is no standardization of the AMD staging in the patients analyzed in the majority of the abovementioned studies. The heterogeneity of AMD suggests that, perhaps, the effects of statins vary by stage, reducing the development of drusen at the onset of AMD, including having a further anti-inflammatory effect on late disease [1]. In addition, genetic analysis should also be performed to understand whether the genotype is capable of influencing a response to this therapeutic option.
Another relevant point is the lack of standardization of statin dosages or individual lipophilicity for each class of drug, considering the combined registration of the use of low-potency and high-potency statin causes a methodological bias [2]. Statin use may only be effective if lipid levels are reduced. It is still unclear how much reduction is required and of which lipid sub-species in order to impact the natural history of AMD, if any effect exists [3]. Finally, other weaknesses in the study design that might lead to insufficient evidence that statins could prevent AMD include the following: short duration of follow-up, lack of data from participants, insufficient sample power to detect significant differences between treatment groups, and the fact that people taking statins, in epidemiological studies, are more likely to have other confounders that can not be fully adjusted, such as high-risk cardiovascular disease and a large number of other medications. Resolving these issues in future studies should lead to a reliable and definitive ruling on the efficacy of statins in AMD.
REFERENCES
1. Ma, L., et al., The association between statin use and risk of age-related macular degeneration. Sci Rep, 2015. 5: p. 18280.
2. Miller, J.W., S. Bagheri, and D.G. Vavvas, Advances in Age-related Macular Degeneration Understanding and Therapy. US Ophthalmic Rev, 2017. 10(2): p. 119-130.
3. Waugh, N., et al., Treatments for dry age-related macular degeneration and Stargardt disease: a systematic review. Health Technol Assess, 2018. 22(27): p. 1-168.
Reviewer 2 Report
The present review discusses the role of statins in age-related macular degeneration (AMD). It is an interesting idea to discuss but the manuscript can be accepted after taking the following changes.
a) Title of the paper should be changed. This review concludes that there is no clear evidence that could suggest that the statin are useful for prevention or delay the onset of AMD.
b) Please check for typos for e.g. heading and subheading numbers
c) The introduction needs significant re-organization for the ease of understanding. For example, it starts with clinical stats of the disease, then talks about VEGF as a druggable target, then comes back to clinical stats, talks about disease diagnostics etc. Authors should rearrange or re-write their introduction to create some connectivity for the ease of understanding.
d) Kindly elaborate on Pathology of AMD, Diagnosis of AMD and Conventional Treatments of AMD as the separate paragraphs. Briefly discuss, if the new class of drug like statin would be of any importance (for the treatment of AMD) compared to existing therapeutic regime?
e) Authors are requested to explain their positive and negative clinical literature data for understanding the correlation between statin and AMD treatment in terms of doses, clinical and pharmacological outcomes of patients. They should discuss why they think that the existing study designs were unable to provide a clear understanding of the possible use of statin in the treatment of AMD?
Author Response
a) The title of the paper was changed to “The question of a role for statins in age-related macular degeneration”.
b) The authors addressed these concerns by correcting the heading and subheading numbers, among other typos.
c) As suggested, the order of the paragraphs of the introduction has been re-organized in order to create some connectivity for easier understanding.
d) The text has been restructured so that AMD's pathology, diagnosis and conventional treatments are now in separate paragraphs. In addition, considering the inherent limitations in currently available therapeutic modalities, certainly a new class of drugs, such as statins, that would be able to expand the therapeutic arsenal would be of great clinical relevance for the treatment of AMD. One sentence was included in the session "Statin and AMD" to clarify this issue.
e) The positive and negative clinical data in the literature was better analyzed in terms of doses and patient outcomes, both, clinical and pharmacological. Moreover, a "Limitation" section was included at the end of the main text, explaining why the authors believe that existing literature designs were unable to support or reject the hypothesis of possible statin use in the treatment of AMD.
Round 2
Reviewer 2 Report
Present edited version of the manuscript shows significant improvement over the previously submitted work. I would recommend to accept this manuscript after considering following minor additions 1) If possible authors are requested to create the table with name Statins (atorvastatin, simvastatin, etc) which were explored for the treatment of AMD under non clinical or clinical scenario with references. If the sufficient data is unavailable then do they think that this could also be the reason behind primitive correlation behind use of statins and AMD? 2) Under subsection 2. Although it clearly explains the importance of Statins in AMDs. But it will be interesting to discuss following "https://www.ncbi.nlm.nih.gov/pmc/articles/PMC5443823/"
Author Response
We are thankful again to the reviewers for their insightful comments, which we hope that we have adequately addressed. Our research team is positive that changes will improve the scientific quality of the manuscript.
1) Under subsection 2. Although it clearly explains the importance of Statins in AMDs. But it will be interesting to discuss following "https://www.ncbi.nlm.nih.gov/pmc/articles/PMC5443823/"
Answer. As suggested, we added a new paragraph (below) under subsection 2 discussing the article entitled “Atorvastatin Promotes Phagocytosis and Attenuates Pro Inflammatory Response in Human Retinal Pigment Epithelial Cells”. The paragraph is highlighted in red.
“Tian et al showed that high doses of atorvastatin preserved the phagocytic function of human RPE cells by increasing cell membrane fluidity. Interestingly, atorvastatin not only increase the baseline phagocytic function of RPE cells, but also preserved the phagocytic function impaired by cholesterol crystals. Similar effects were observed in assessing lovastatin and simvastatin. Furthermore, atorvastatin also exhibited anti-inflammatory properties, blocking the induction of IL-6 and IL-8 interleukins triggered by pathological stimuli relevant to AMD. Both characteristics reinforce the role of statins as a potentially effective drug in the prevention and treatment of AMD by protecting RPE cells from the impairment of phagocytic function and the inflammatory effects induced by cholesterol crystals (Tian, et al 2017)”
2) If possible authors are requested to create the table with name Statins (atorvastatin, simvastatin, etc) which were explored for the treatment of AMD under non clinical or clinical scenario with references. If the sufficient data is unavailable then do they think that this could also be the reason behind primitive correlation behind use of statins and AMD?
Answer. The requested table was included in the main text